# Cross-sectional study using primary care and cancer registration data to investigate patients with cancer presenting with non-specific symptoms

Clare Pearson ![ORCID] ,[1,2] Veronique Poirier,[1] Karen Fitzgerald,[1] Greg Rubin,[3] Willie Hamilton[4]

¹Policy and Information, Cancer Research UK, London, UK
²National Cancer Registration and Analysis Service, Public Health England, London, UK
³Institute of Health and Society, Newcastle University, Royal Victoria Infirmary, Newcastle upon Tyne, UK
⁴Primary Care, Medical School, University of Exeter, Exeter, UK

**Correspondence to**
Clare Pearson;
clare.pearson@cancer.org.uk

## ABSTRACT

**Introduction** Patients presenting to primary care with site-specific alarm symptoms can be referred onto urgent suspected cancer pathways, whereas those with non-specific symptoms currently have no dedicated referral routes leading to delays in cancer diagnosis and poorer outcomes. Pilot Multidisciplinary Diagnostic Centres (MDCs) provide a referral route for such patients in England.

**Objectives** This work aimed to use linked primary care and cancer registration data to describe diagnostic pathways for patients similar to those being referred into MDCs and compare them to patients presenting with more specific symptoms.

**Methods** This cross-sectional study linked primary care data from the National Cancer Diagnosis Audit (NCDA) to national cancer registration and Route to Diagnosis records. Patient symptoms recorded in the NCDA were used to allocate patients to one of two groups - those presenting with symptoms mirroring referral criteria of MDCs (non-specific but concerning symptoms (NSCS)) and those with at least one site-specific alarm symptom (non-NSCS). Descriptive analyses compared the two groups and regression analysis by group investigated associations with long primary care intervals (PCIs).

**Results** Patients with NSCS were more likely to be diagnosed at later stage (32% stage 4, compared with 21% in non-NSCS) and via an emergency presentation (34% vs 16%). These patients also had more multiple pre-referral general practitioner consultations (59% vs 43%) and primary care-led diagnostics (blood tests: 57% vs 35%). Patients with NSCS had higher odds of having longer PCIs (adjusted OR: 1.24 (1.11 to 1.36)). Patients with lung and urological cancers also had higher odds of longer PCIs overall and in both groups.

**Conclusions** Differences in the diagnostic pathway show that patients with symptoms mirroring the MDC referral criteria could benefit from a new referral pathway.

## INTRODUCTION

Earlier detection of cancer improves clinical outcomes and quality of life for patients with cancer, through improved treatment options and increased likelihood of survival. Patients in England who present to their general practitioner (GP) with site-specific 'alarm' symptoms are likely to be referred via an urgent pathway under the 'Suspected Cancer: recognition and referral' guidelines from the National Institute for Health and Care Excellence (NICE)[1] and are subsequently monitored within the Cancer Waiting Times timeframes.[2] This 'Two Week Wait' (TWW) diagnostic route refers to the 14-day target between referral and seeing a specialist.[3] However, previous studies have identified large proportions of patients with cancer diagnosed without having alarm symptoms in general practice in Denmark (52%),[4] with similar proportions in the UK[5] and higher rates in Norway (60%).[6 7]

### Strengths and limitations of this study

► The key strength of this study is the linkage of cancer registrations to primary care data enhancing our understanding of the diagnostic pathway for patients with cancer.

► Comparing different groups of patients based on their presenting symptoms in primary care enables a focus on patients with symptoms similar to those being referred into England's pilot Multidisciplinary Diagnostic Centres (MDCs) who do not currently have an urgent referral pathway.

► A limitation of this study is that it does not compare patients presenting with symptoms that would trigger an urgent referral under the National Institute for Health and Care Excellence (NICE) guidelines to those who experience non-urgent referral symptoms; instead, it mirrors the referral criteria of the MDCs to describe diagnostic pathways for these patients.

► Symptoms recorded in our study in primary care relied on accurate reporting.

► Not all MDC referral criteria could be included in our study, notably general practitioner intuition/patient concern which were not recordable in the data.

In England, for patients who present without such site-specific alarm symptoms warranting a TWW referral, it can be challenging for GPs to select the most appropriate referral pathway. Examples of such non-specific but concerning symptoms (NSCS) include unexplained weight loss, fatigue and some types of abdominal pain.[8] Patients presenting with NSCS could therefore experience repeated referrals to different secondary care departments before a cancer diagnosis is confirmed. Overall, this could lead to unstructured and prolonged diagnostic pathways, which could negatively impact on outcomes, such as poorer survival,[9] later stage[9] or worse patient experience.[10] Patients with less alarm/specific symptoms are more likely to be diagnosed via emergency presentations,[11] which is itself associated with poorer outcomes.[12] If diagnostic pathways for such patients could be improved, there is an opportunity to improve outcomes.

A potential solution for these patients has been recently trialled and evaluated by the Accelerate, Coordinate and Evaluate (ACE) Programme, which is a joint initiative between Cancer Research UK, Macmillan Cancer Support and NHS England. The programme aims to achieve earlier cancer diagnosis. Wave 2 of the ACE Programme is evaluating whether Multidisciplinary Diagnostic Centres (MDCs) can support earlier and faster diagnosis of cancers/non-cancer conditions for patients with no clear urgent diagnostic pathway using a symptom-based approach to streamline diagnostic pathways for such patients. The MDCs aim to provide comprehensive diagnostics under the care of the same team to provide a more rapid diagnosis of cancer and other conditions.[13] Similar programmes have been implemented in Denmark[4 14] and Sweden.[15] There is currently little evidence of whether unmet need exists for patients in England likely to be referred into the MDCs and how their diagnostic pathways compare with patients presenting with more specific symptoms.

The National Cancer Diagnosis Audit (NCDA)[16] provides an opportunity to explore rich primary care data for patients with cancer diagnosed in 2014 and when linked to cancer registrations and other health datasets held at the National Cancer Registration and Analysis Service (NCRAS), Public Health England, it can build a more complete picture of diagnostic pathways. Symptoms at presentation to the GP are recorded in the NCDA enabling comparison of groups of patients presenting with different kinds of symptoms.

The aim of this particular study was to use linked primary care data to provide understanding of any unmet need for patients similar to those being referred into the MDCs and to compare them with patients presenting with more specific symptoms, specifically focusing on factors which could lead to poorer outcomes such as stage and route to diagnosis. In addition, primary care intervals (PCIs) for both groups of patients were compared.

## METHODS

### Datasets

The NCDA was conducted using primary care data submitted from participating GP surgeries on a voluntary basis. Cancer registrations from 2014 in England were sent to these surgeries where primary care information was collated to create the NCDA. This included dates of presentation and referral, symptoms at presentation, primary care-led investigations and many others.[16] Eighty-three per cent of participating surgeries completed over 95% of patient NCDA data with over 17 000 cancers submitted in total. The Routes to Diagnosis (RtD) dataset is generated at NCRAS, using several linked health datasets to determine the most likely diagnostic route,[17] including emergency presentations, inpatient, TWW and routine GP referrals.

### Data linkage

The NCDA and RtD datasets were linked with cancer registration data at tumour level using tumour ID. Where a patient had multiple cancers (n=385), GPs were instructed to enter the same demographic and patient details, while submitting symptoms, investigations and interval data for each cancer separately.

### Allocation to symptom groups

There were 84 distinct symptoms listed in the NCDA. To reflect patients being referred into the MDCs, the patients in the linked dataset were allocated to one of two groups depending on the symptoms at presentation to the GP within the NCDA dataset. The symptoms used to allocate patients were derived from the combined referral criteria to the MDCs[18] and an additional common presenting symptom (Bowel habit change) of the MDCs. These symptoms are listed in table 1.

To be allocated to the NSCS group, patients could only have symptom(s) listed in table 1. If a patient had symptom(s) listed in table 1 but in addition had one or more other symptom, usually triggering a TWW referral (eg, a lump, bleeding), they were allocated to the non-NSCS group. Patients with unknown symptoms in the NCDA were excluded; reasons for this could be due to screening (where GPs could not enter symptoms) or that the symptoms were not known to the GP.

The PCI was defined as the time from first relevant presentation to the GP to when the patient is referred into secondary care.[19] The first presentation date in the NCDA was completed as the date when the patient first presented with symptoms ultimately attributed by the GP to the diagnosis of cancer.[16]

Cancer sites were categorised into the Cancer Waiting Times site specific grouping[20] depending on their International Classification of Disease v10 code. They are listed in the online supplementary information, table S1.

### Exclusions

Patients with no symptom information could not be allocated to one of the symptom-based groups and

**Table 1** List of non-specific but concerning symptoms—Multidisciplinary Diagnostic Centres (MDC) referral criteria and a common presenting symptom

| Symptom | Notes |
|---|---|
| Distention | |
| Pallor | |
| Abdominal pain (upper, lower, NOS*) | *NOS (not otherwise specified) |
| Bowel habit change | |
| Constipation | |
| Diarrhoea | |
| Nausea and/or vomiting | |
| Fatigue | |
| Weight loss | |
| Back pain | |
| New onset diabetes | |
| Lymphadenopathy | (generalised and localised) |
| Deep vein thrombosis | |
| Loss of appetite | |
| Chest pain | |
| Chest infection | |
| Jaundice† | † For local reasons, this specific symptom was a referral symptom in one MDC project |

were therefore excluded (n=3844) (figure 1). We also excluded cancers diagnosed via death certificate (n=13) or screening (n=14) in the RtD. Patients with PCIs of a negative value (n=218) or over 730 days (n=80) were excluded as in previous methodology.[16 21]

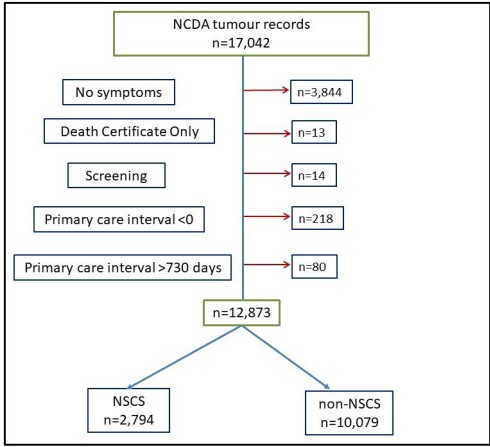

**Figure 1** Data exclusions and allocation to symptom-based analysis groups.
**NCDA, National Cancer Diagnosis Audit; NSCS, non-specific but concerning symptoms.**

## Statistical analysis

Comorbidities were recorded in the NCDA and categorised by patient in our analysis. Frequencies and proportions of patients by different sociodemographic and disease characteristics were described in the NSCS and non-NSCS groups. Differences between proportions in the two groups by characteristic were assessed by $\chi^2$ tests. The PCIs were also described by characteristic and NSCS/non-NSCS group. Based on the distribution of the PCI and clinical advice, the interval was divided into less than and including 28 days and over 28 days for regression analysis to denote a longer PCI. We used multivariable logistic regression with longer PCI as the outcome variable with sociodemographic and disease characteristics as explanatory variables (age group, sex, comorbidities, deprivation, route to diagnosis, cancer site and stage). The regression analysis was also stratified by NSCS/non-NSCS group. Analyses were conducted in Stata V.15.1.

### Patient and public involvement

Data for this study are based on information collected by the NHS. Patients and the public were not involved in the development of this study.

## RESULTS

There were 17 042 cancers records in the NCDA linked to cancer registration data. Following exclusions, 2794 (22% of remaining NCDA cohort) patients had only MDC symptoms recorded and were allocated to the NSCS group. 10 079 (78%) patients had at least one non-NSCS symptom and were therefore allocated to the non-NSCS group (figure 1). Table 2 shows frequencies of patients in each of the NSCS and non-NSCS groups by sociodemographic and disease characteristics along with the PCI for each group and characteristic. Table 3 describes pathway and disease characteristics and corresponding PCIs.

The NSCS group were older with the median (IQR) age in the NSCS group of 72 years (63–81) and 69 (58–78) in non-NSCS. There were higher proportions of the NSCS group in the two oldest age groups. All factors in table 2 (apart from sex) show that there were significantly different proportions in NSCS compared with non-NSCS. Higher proportions of the NSCS group resided in more deprived areas and had more comorbidities. There were higher proportions of the NSCS group diagnosed via emergency presentation and lower proportions in the TWW diagnostic route. Excluding unknown stage, 33% of NSCS group were diagnosed at stages 1 or 2, compared with 55% in the non-NSCS group, with correspondingly higher proportions of NSCS patients diagnosed at late stage (excluding unknown: stages 3 and 4: NSCS 67%, non-NSCS 45%).

PCIs were consistently longer in the NSCS group in all categories of characteristics and at all levels with a wider IQR, with only a couple of exceptions. Patients diagnosed via the emergency presentation route had the same PCI for both groups. Patients under 25 years in non-NSCS had

**Table 2**  Characteristics of non-specific but concerning symptoms (NSCS) and non-NSCS patients and primary care intervals (median and IQR)

| | Frequencies | | | Primary care interval (days) | |
|---|---|---|---|---|---|
| | **NSCS** | **Non-NSCS** | | **NSCS** | **Non-NSCS** |
| | **n (%)** | **n (%)** | **P value*** | **Median (IQR)** | **Median (IQR)** |
| Total | 2794 (21.7) | 10079 (78.3) | | 12 (1–39) | 3 (0–23) |
| Age group | | | | | |
| 0–24 | 33 (1.18) | 123 (1.22) | <0.001 | 0 (0–47.5) | 5 (0–29) |
| 25–44 | 105 (3.76) | 712 (7.06) | | 13 (0–45) | 0 (0–15) |
| 45–59 | 387 (13.85) | 1865 (18.50) | | 13 (0–42) | 1 (0–19) |
| 60–69 | 645 (23.09) | 2439 (24.20) | | 14 (1–41) | 5 (0–28) |
| 70–79 | 827 (29.60) | 2759 (27.37) | | 13 (1–42) | 4 (0–24) |
| 80+ | 797 (28.53) | 2181 (21.64) | | 9 (1–31) | 3 (0–22) |
| Sex | | | | | |
| Male | 1410 (50.47) | 5191 (51.50) | 0.331 | 11 (0–36) | 7 (0–29) |
| Female | 1384 (49.53) | 4888 (48.50) | | 13 (1–43) | 0 (0–15) |
| Deprivation quintile | | | | | |
| 1—least deprived | 557 (19.94) | 2227 (22.10) | 0.050 | 11 (1–36) | 3 (0–23) |
| 2 | 599 (21.44) | 2182 (21.65) | | 13 (1–41) | 3 (0–26) |
| 3 | 595 (21.30) | 2182 (21.65) | | 11.5 (1–35) | 2 (0–23) |
| 4 | 567 (20.29) | 1900 (18.85) | | 11 (0–42.5) | 2 (0–21) |
| 5—most deprived | 476 (17.04) | 1588 (15.76) | | 13 (0–41.5) | 3 (0–24) |
| Comorbidities | | | | | |
| 0 | 661 (23.66) | 2787 (27.65) | <0.001 | 9 (0–37) | 1 (0–21) |
| 1 | 853 (30.53) | 2993 (29.70) | | 13 (1–43) | 3 (0–25) |
| 2 | 678 (24.27) | 2289 (22.71) | | 13 (1–39) | 3 (0–23) |
| 3+ | 602 (21.55) | 2010 (19.94) | | 13 (1–39) | 4 (0–28) |
| Diagnostic route | | | | | |
| Emergency presentation | 949 (33.97) | 1623 (16.10) | <0.001 | 5 (0–27) | 5 (0–26) |
| GP referral | 681 (24.37) | 2509 (24.89) | | 14 (1–44) | 7 (0–31) |
| Two Week Wait | 802 (28.70) | 4782 (47.45) | | 14 (3–42) | 1 (0–17) |
| Inpatient elective | 60 (2.15) | 171 (1.70) | | 9.5 (0–45) | 5 (0–48) |
| Outpatient | 235 (8.41) | 724 (7.18) | | 12 (0–45) | 5 (0–29) |
| Unknown | 67 (2.40) | 270 (2.68) | | 10.5 (0–27) | 8 (0–45) |
| Stage | | | | | |
| 1 | 290 (10.38) | 2573 (25.53) | <0.001 | 13 (0–42) | 0 (0–15) |
| 2 | 357 (12.78) | 1828 (18.14) | | 10 (0–39) | 0 (0–14) |
| 3 | 444 (15.89) | 1516 (15.04) | | 12 (1–38) | 5 (0–28) |
| 4 | 897 (32.10) | 2094 (20.78) | | 14 (1–42) | 8 (0–35) |
| Unknown/other | 806 (28.85) | 2068 (20.52) | | 10 (0.5–37) | 5 (0–28) |

*Significance test of proportions in NSCS/non-NSCS by characteristic.
GP, general practitioner.

longer PCIs than the NSCS group. PCIs were longer for every stage of disease at diagnosis in the NSCS group.

There were higher proportions of patients with multiple consultations and more primary care-led investigations (apart from urinary investigations) in the NSCS group,

there was also a higher proportion where the GP felt that there was an avoidable delay to their diagnosis in this group (table 3). Patients could have more than one investigation, so significance testing was not undertaken on this element in table 3; however, the other characteristics

**Table 3** Pathway characteristics of non-specific but concerning symptoms (NSCS) and non-NSCS patients and primary care intervals (median and IQR)

| | NSCS n (%) | Non-NSCS n (%) | P value* | Primary care interval (days) | |
| | | | | NSCS | Non-NSCS |
| | | | | Median (IQR) | Median (IQR) |
|---|---|---|---|---|---|
| **Number of consultations before referral†** | | | | | |
| 1 | 792 (28.35) | 4362 (43.28) | <0.001 | 0 (0–4) | 0 (0–1) |
| 2 | 742 (26.35) | 2197 (21.80) | | 12 (5–25) | 14 (6–30) |
| 3 | 356 (12.74) | 882 (8.75) | | 28 (11–59) | 30 (14–58.5) |
| 4 | 196 (7.02) | 460 (4.56) | | 40 (21–73) | 42 (20–88) |
| 5+ | 353 (12.63) | 761 (7.55) | | 58 (33–134) | 76 (33–148) |
| **Primary care-led investigations** | | | | | |
| Blood tests | 1589 (56.87) | 3510 (34.82) | | 15 (4–45) | 14 (3–41) |
| Urinary | 21 (0.75) | 175 (1.74) | | 35 (8–69) | 14 (5–38) |
| Imaging | 930 (33.29) | 2176 (21.59) | | 23 (7–55) | 21 (6–50.5) |
| Imaging: X-ray | 393 (14.07) | 1330 (13.20) | | 21 (6–55) | 18 (5–50) |
| Imaging: CT | 123 (4.40) | 226 (2.24) | | 27 (7–53.5) | 35 (16–76.5) |
| Imaging: Ultrasound | 513 (18.36) | 818 (8.12) | | 28 (10–58) | 24 (8–51) |
| Imaging: MRI | 32 (1.15) | 61 (0.61) | | 35 (9–60) | 33.5 (11–98) |
| Endoscopy | 95 (3.40) | 152 (1.51) | | 38 (6–94) | 11.5 (0–50) |
| Endoscopy: Upper GI‡ | 52 (1.86) | 102 (1.01) | | 41 (7–95) | 14 (0–50) |
| Endoscopy: Colon | 45 (1.61) | 44 (0.44) | | 42 (5–108) | 7 (0–73) |
| Other | 314 (11.24) | 904 (8.97) | | 20 (6–50) | 17 (4–49) |
| None | 712 (25.48) | 4544 (45.08) | | 0 (0–12) | 0 (0–2) |
| **Avoidable delays?** | | | | | |
| Yes | 738 (26.41) | 2215 (21.98) | <0.001 | 30 (6–85) | 21 (0–70) |
| No | 1825 (65.32) | 7256 (71.99) | | 7 (0–25) | 1 (0–14) |
| Unknown | 231 (8.27) | 608 (6.03) | | 15 (0–42) | 6 (0–36) |
| **Cancer site** | | | | | |
| Brain and CNS§ | 16 (0.57) | 208 (2.06) | <0.001 | 64 (35–99) | 3 (0–16) |
| Breast | 17 (0.61) | 1614 (16.01) | | 15 (3–33) | 0 (0–0) |
| Colorectal | 804 (28.78) | 817 (8.11) | | 8 (0–36) | 3 (0–25) |
| Gynaecology | 179 (6.41) | 634 (6.29) | | 14 (3–35) | 1 (0–19) |
| Haematology | 338 (12.10) | 782 (7.76) | | 12 (1–38) | 10 (1–35) |
| Head and neck | 12 (0.43) | 527 (5.23) | | 18 (0–45) | 3 (0–28) |
| Lung | 423 (15.14) | 1427 (14.16) | | 14 (1–45) | 14 (2–45) |
| Sarcoma | 28 (1.00) | 121 (1.20) | | 12 (1–43) | 12.5 (0–46.5) |
| Skin | <5 (<1%) | 742 (7.36) | | N/A | 0 (0–2) |
| Upper GI‡ | 551 (19.72) | 750 (7.44) | | 10 (1–36) | 5 (0–33) |
| Urology | 279 (9.99) | 2256 (22.38) | | 15 (3–43) | 10 (1–31) |
| Other | 145 (5.19) | 201 (1.99) | | 9 (0–35) | 6 (0–32) |

*Significance test of proportions in NSCS/non-NSCS by characteristic.
†Excluding 0 consultations.
‡Upper gastrointestinal (GI).
§Central nervous system (CNS).

in table 3 had statistically significantly different proportions in NSCS/non-NSCS with all p values<0.001. There were higher proportions of breast, head and neck, brain and CNS and urological cancers in the non-NSCS group. Median PCIs were the same in both groups for lung cancer, similar for sarcoma, but longer in the NSCS group for all other groupings.

The PCI was longer in the non-NSCS group for patients presenting more than twice before referral. There was variation in PCI by group for investigations, with longer

intervals in the NSCS group for most tests, the exception being CT. PCIs were longer in those with an avoidable delay in the NSCS group and longer in both groups where there was an avoidable delay to diagnosis.

Table S2 (online supplementary information) shows unadjusted and adjusted ORs of having a long PCI for the entire NCDA cohort (n=12873) and stratified by NSCS (n=2974)/non-NSCS (n=10079) group. In the entire NCDA cohort, after adjustment for age, sex, deprivation, comorbidities, route, stage and site, being in the NSCS group was associated with having a longer PCI (adjusted OR (95% CIs 1.24 (1.12–1.36)). Compared with TWW, all other routes had higher odds of longer intervals. Higher comorbidity scores were also associated with longer PCIs. Compared with colorectal cancers, patients with haematological, lung, sarcoma, brain and CNS and urological cancers were more likely to have longer PCIs. When stratified by NSCS/non-NSCS, comorbidity remained significantly associated with longer PCIs for all scores in NSCS and only in the highest category in the non-NSCS group. Associations by diagnostic route remained significant with a similar pattern for both groups, with lower odds in the NSCS group. Breast cancers had lower odds of having longer PCI in only the non-NSCS group and only lung, brain and CNS and urological cancers had higher odds in both groups of having longer PCIs compared with colorectal cancer.

## DISCUSSION
### Summary of main findings
This large study used existing data to examine patients with cancer who could have been eligible for referral to an MDC. It showed clear differences in such patients and those eligible for urgent suspected cancer referral, whereby the former experienced longer PCIs, had more primary care interactions, were more likely to be diagnosed at later stage and via emergency presentation.

### Strengths and limitations
The use of primary care data linked with cancer registrations enables a detailed picture of the diagnostic pathway for patients with cancer. Symptoms recorded in the NCDA provide a basis for examining different groups of patients.

The allocation into the NSCS and non-NSCS groups by symptom is a proxy for distinguishing between alarm and non-alarm symptoms in the NICE referral guidelines. An analysis of symptom groups with a true separation between alarm symptoms warranting a referral onto an urgent referral pathway and vague symptoms which do not would be very difficult. This is especially the case for symptoms where there are more than one recommendation depending on other symptoms and patient characteristics (such as appetite loss, with five different recommendations and weight loss with 13 recommendations). Such analysis would also require patient characteristic information which is not all available in the NCDA-linked data. This study, however, focused on the MDC referral criteria and common presenting symptoms recorded at MDCs. Indeed, the aim of this work was to provide evidence of the possible diagnostic problems facing patients similar to those potentially eligible to go through the MDCs.

Not all MDC referral criteria were recorded in the NCDA, including GP intuition or patient/family concern, though it is unlikely that the inclusion of this would have changed the allocation to symptom groups. Additionally, not all of the non-specific symptoms in our allocation list are truly low risk, with jaundice being the most debatable, instead being an alarm symptom for particular cancers, and it is genuinely high risk for pancreatic cancer, though much lower for other cancers.[22 23] Symptoms recorded in the NCDA were not necessarily complete and relied on accurate recording in primary care systems for those completing the NCDA to extract.[16]

### Comparisons with the literature
Previous work using the NCDA has shown significant variation in the patient interval (symptom onset to presentation) by different abdominal symptoms,[24] and cohort studies have shown similar variation of different diagnostic intervals by symptoms for patients with colorectal,[25] lung[26] and pancreatic cancer.[27] Other studies using linked primary care data have found longer diagnostic intervals (from symptom presentation to diagnosis) for those presenting with non-alarm symptoms[5 28 29] when compared with alarm symptoms patients. A previous study examining patients with lung cancer found longer PCIs for patients presenting with vague symptoms.[30]

Previous work on PCIs have shown variation in these intervals by cancer site,[16 31 32] yet only one has focused on different symptom profiles of patients with lung cancer.[30]

Our study adds to this body of literature by examining the diagnostic pathways of patients diagnosed in England with a wide range of cancers presenting with NSCS.

### Interpretation and implications
The lack of specific referral pathways for patients who present with non-specific symptoms is well described.[33 34] Our work explains the problems facing patients who presented with non-specific symptoms, similar to those to be referred into the MDCs, with longer PCIs more primary care interactions. The higher proportion of late stage disease in those presenting with NSCS may relate to the passing of time until the symptoms became more pronounced, leading to a cancer diagnosis at a later stage of disease, possibly via an emergency, which we show that NSCS patients are also more likely to experience. These patients have longer time intervals before referral to secondary care indicating the lack of clear referral route onto a specific urgent cancer referral pathway.

The association between certain sites (lung, urology) and longer PCIs was evident in both groups and overall, probably due, in part, to presenting symptoms.

We have demonstrated patients presenting with NSCS who would fulfil the criteria for MDC referral take longer to reach a diagnosis than those likely to be referred on an urgent suspected cancer pathway. They also have higher proportions of late stage/emergency presentations. This study does not show that MDCs can expedite diagnosis, but indicates the problems facing patients diagnosed with cancer who present with non-specific symptoms, for which MDCs may be the answer. The results of the MDC evaluations will be published separately.

## CONCLUSION

Using national linked data, we have demonstrated that patients presenting with NSCS experienced longer time intervals before diagnosis, were more likely to be diagnosed via an emergency and at a later stage of disease, all of which are associated with poorer outcomes. An alternative diagnostic referral pathway for these patients should therefore be considered.

**Acknowledgements** The work was undertaken as part of the CRUK-PHE partnership and the ACE Programme, which is CRUK, Macmillan cancer support and NHS England supported. The authors would also like to thank Ruth Swann and Jess Fraser (CRUK-PHE partnership), Dave Chapman (ACE Programme, CRUK) and the National Cancer Diagnosis Audit. Data for this study are based on patient-level information collected by the NHS, as part of the care and support of patients with cancer. The data are collated, maintained and quality assured by the National Cancer Registration and Analysis Service, which is part of Public Health England

**Contributors** Conception and design of the work: CP, VP, KF and WH. Analysis of the data: CP with input from VP. Contributions to the interpretation of the findings: all authors. All authors contributed to drafting the manuscript or revising it critically for important intellectual content and approved the final version submitted. All authors have agreed to be accountable for all aspects of the work.

**Funding** WH is co-PI and GR steering group chair of CanTest, a Cancer Research UK Catalyst Award [C8640/A23385].

**Competing interests** None declared.

**Patient consent for publication** Obtained.

**Ethics approval** This study was exempt from gaining individual consent having obtained Section 251 approval from the UK Patient Advisory Group (PIAG) (now the Confidentiality Advisory Group, CAG), under Section 251 of the NHS Act 2006 (PIAG 03(a)/2001).

**Provenance and peer review** Not commissioned; externally peer reviewed.

**Data availability statement** Data are available upon reasonable request. Enquiries for data access can be made to Public Health England's Office for Data Release (odr@phe.gov.uk).

**ORCID iD**
Clare Pearson http://orcid.org/0000-0002-1941-4735

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
