## [Reviewer comments · BMJ Open]

ARTICLE DETAILS

TITLE (PROVISIONAL)	Cross-sectional study using primary care and cancer registration data to investigate cancer patients presenting with non-specific symptoms
AUTHORS	Pearson, Clare; Poirier, Veronique; Fitzgerald, Karen; Rubin, Greg; Hamilton, Willie

VERSION 1 – REVIEW

REVIEWER	Rebecca Bergin Cancer Council Victoria Australia
REVIEW RETURNED	19-Aug-2019

GENERAL COMMENTS	BMJ Open Bmjopen-2019-033008 Reviewer: Rebecca Bergin, 19/8/2019 Title: Cross-sectional study using primary care and cancer registration data to investigate cancer patients presenting with non-specific symptoms. This study examines factors associated with cancer patients who present with non-specific but concerning symptoms (NSCS). The categorisation of these non-specific symptoms is based on referral criteria as well as commonly reported symptoms in patients taking part in a program currently being tested, Multidisciplinary Diagnostic Centres (MDCs), in England. To examine differences between those with NSCS and non-NSCS, the work uses a very large primary care audit from 2014 linked to diagnostic route information and is an interesting use of data that could inform ongoing programs to improve timely cancer diagnosis. I have some comments and queries on the manuscript: 1. The aim in the abstract is to 'describe the problems' facing patients similar to those referred to MDCs, while the aim in the introduction is 'to provide an understanding of unmet need' for these patients. Both aims are vague. The reference to 'unmet need' is not clear what kind of needs are of interest. The conclusions also do not refer to 'problems' or 'unmet needs' faced by patients. Because of the lack of clarity around the stated aim, I am unsure whether the study achieves this aim.2. It was surprising that, although the analysis uses MDC criteria to categorise patients, there is very little discussion about the implications of study findings for the MDC program.3. The study sample appears to include all cancer types. As noted in the discussion, cancer type is probably important in explaining why there are differences between the NSCS and non-NSCS groups, such as more women which may be driven by breast cancer patients in the NSCS group. This may also explain the faster primary care interval observed for this group. Type of cancer
---

	likely also impacts the type of tests that GPs order. Cancer type is not adjusted for in Table 4 regression analyses. I would suggest including a breakdown by cancer type for each group in table 1, and further reflection of the possible impact of cancer type in the discussion. 4. The methods are a little sparse. a. It would be useful to know who completed the NCDA audit data – was it GP or were electronic primary care records extracted for relevant cases? Also, what was the response rate for participation in the NCDA audit? b. How successful, and accurate, was the linkage between NCDA and RtD datasets? What was done with cases with multiple cancer diagnoses? c. How was the date of first relevant presentation to a GP determined? d. It was a little unclear how the MDC referral criteria and common presenting symptoms of MDCs were determined – what were the common presenting symptoms in MDCs vs referral criteria? Which MDCs were data taken to determine common symptoms / how many patients was this for? How was ‘common’ defined? Has this data been published? 5. It appears that some significant effects are not highlighted in Table 4 (e.g. unadjusted OR for age 25-44, and 45-59; non-NSCS unadjusted OR for age 25-44, 45-59; unadjusted OR stage 2; non-NSCS unadjusted OR stage 2). 6. The abstract refers to the MDC-group vs non-MDC group, but paper uses terms NSCS and non-NSCS; consistency would be useful. 7. Sample size for regression models are not stated and abbreviations not defined in results tables.
--	---

REVIEWER	John Primrose University of Southampton, UK
REVIEW RETURNED	05-Sep-2019

GENERAL COMMENTS	Patients with NICE describe alarm symptoms for cancer are investigated via a rapid access (TWW) pathway. However, the majority of such patients do not have cancer and there is a serious question as to whether this pathway is disadvantaging patients with less specific symptoms. With some cancers, it is clear that a majority of cancer presentations are from other than the TWW patients. How to manage this group is a significant issue in cancer care as they are clearly more numerous. In this study the authors try and use cancer data linked to primary care data to establish the characteristics of a cohort similar to those who may be seen in one of the experimental Multidisciplinary Diagnostic Centres (MDCs) in which the authors have an interest. The results reveal that this cohort with non-specific symptoms were likely to be older, more deprived, more comorbid and have higher stage disease compared to the “other group”. The results as far as they go are of some interest although largely predictable. It describes the characteristics of a group who had ultimately a diagnosis of cancer but not the denominator. The denominator is critical if the utility of MDCs is to be assessed. There are some specific problems with this paper.
---

	 1. It is very difficult to follow bearing in mind the relatively simple (I think) message. It is also over long. 2. The numerous similar abbreviations do little enhance it accessibly. 3. Although I have read it many time I still don't know if non-NSCS is the TWW group. If it is please say so. If it isn't could the authors explain. 4. There is a large amount of data in the tables. I doubt anybody apart from a hard-core epidemiologist will actually take an interest in these data and perhaps much could be a as supplement. 5. The important data could be presented more simply. 6. Although the limitations of the study are presented the reality is this simply describes characteristic of patients who turned out to have cancer but did not have (I think) TWW symptoms and it does not crucially look at outcome. 7. If MDCs were going to be promoted more evidence would be needed. It is as yet unclear that the TWW system has utility. Whilst this is not the aim of the paper some discussion on how it might impact on future studies would be of value 8. Although the authors do comment on jaundice not being in any sense being a non specific symptom, it is so completely ridiculous that these data should probably be excluded.
--	--

VERSION 1 – AUTHOR RESPONSE

Reviewer 1

This study examines factors associated with cancer patients who present with non-specific but concerning symptoms (NSCS). The categorisation of these non-specific symptoms is based on referral criteria as well as commonly reported symptoms in patients taking part in a program currently being tested, Multidisciplinary Diagnostic Centres (MDCs), in England. To examine differences between those with NSCS and non-NSCS, the work uses a very large primary care audit from 2014 linked to diagnostic route information and is an interesting use of data that could inform ongoing programs to improve timely cancer diagnosis. I have some comments and queries on the manuscript:

1. The aim in the abstract is to 'describe the problems' facing patients similar to those referred to MDCs, while the aim in the introduction is 'to provide an understanding of unmet need' for these patients. Both aims are vague. The reference to 'unmet need' is not clear what kind of needs are of interest. The conclusions also do not refer to 'problems' or 'unmet needs' faced by patients. Because of the lack of clarity around the stated aim, I am unsure whether the study achieves this aim.

We thank the reviewer for these comments. We have updated the aims in both the abstract and main body of the paper to better reflect the work undertaken and have removed the vagueness of the 'describing the problems' part of the aims. We have also added to the conclusions in the discussion, so they line up with the stated aims.

The updated abstract reads:

'Objectives

This work aimed to use linked primary care and cancer registration data to describe diagnostic pathways for patients similar to those being referred into MDCs and compare these patients to patients presenting with more specific symptoms.'

The last paragraph of the introduction has been updated to:

'The aim of this particular study was to use linked primary care data to provide understanding of any unmet need for patients similar to those being referred into the MDCs and to compare them with patients presenting with more specific symptoms, specifically focusing on factors which could lead to poorer outcomes such as stage and route to diagnosis. In addition, primary care intervals (PCIs) for both groups of patients will be compared.'

The discussion has been updated to:

'We have demonstrated patients presenting with NSCS who would fulfil the criteria for MDC referral take longer to reach a diagnosis than those likely to be referred on an urgent suspected cancer pathway. They also have higher proportions of late stage/emergency presentations. This study does not show that MDCs can expedite diagnosis, but indicates the problems facing patients diagnosed with cancer who present with non-specific symptoms, for which MDCs may be the answer. The results of the MDC evaluations will be published separately.'

- 2) It was surprising that, although the analysis uses MDC criteria to categorise patients, there is very little discussion about the implications of study findings for the MDC program.

Reviewer 1 makes a valid point here. This work underpins the rationale for the MDCs in terms of highlighting the problems in the diagnostic pathways of patients eligible for the MDCs. In parallel we are performing an evaluation of actual MDCs. The sample size in that evaluation will perform be much smaller. We considered it premature to comment on the implications for MDCs knowing a second paper would be forthcoming. We have stated that the evaluation of the MDCs themselves is being published separately in the discussion (in paragraph above in response to comment 1).

3. The study sample appears to include all cancer types. As noted in the discussion, cancer type is probably important in explaining why there are differences between the NSCS and non-NSCS groups, such as more women which may be driven by breast cancer patients in the NSCS group. This may also explain the faster primary care interval observed for this group. Type of cancer likely also impacts the type of tests that GPs order. Cancer type is not adjusted for in Table 4 regression analyses. I would suggest including a breakdown by cancer type for each group in table 1, and further reflection of the possible impact of cancer type in the discussion.

This is a helpful point. We have added cancer type to table 3, providing descriptors and intervals by broad cancer type, indicating assignment to these broad groups by Table S1 in the Supplementary Information. We have also re-run the regression analysis adjusting for cancer site. Accordingly, we have updated the methods section to read:

'Cancer sites were categorised into the Cancer Waiting Times site specific grouping (20) depending on their ICD10 (International Classification of Disease v10) code. They are listed in table S1 in the Supplementary Information.'

'We used multivariable logistic regression with longer PCI as the outcome variable with socio-demographic and disease characteristics as explanatory variables (age group, sex, comorbidities, deprivation, route to diagnosis, cancer site and stage).'

In the results section we have included this sentence to describe the proportions and intervals by cancer site:

'There were higher proportions of breast, head & neck, brain & CNS and urological cancers in the non-NSCS group. Median PCIs were the same in both groups for lung cancer, similar for sarcoma, but longer in the NSCS group for all other groupings.'

In the results section we have refined the paragraphs describing the regression results to reflect the addition of stage into the models and have also updated the discussion to reflect these changes

4. The methods are a little sparse.

a. It would be useful to know who completed the NCDA audit data – was it GP or were electronic primary care records extracted for relevant cases? Also, what was the response rate for participation in the NCDA audit?

We have updated the methods section to better explain that the NCDA was voluntary. We have also included a sentence about the response rate. The paragraph now reads:

'The National Cancer Diagnosis Audit was conducted using primary care data submitted from participating GP surgeries on a voluntary basis. Cancer registrations from 2014 in England were sent to these surgeries where primary care information was collated to create the NCDA. This included dates of presentation and referral, symptoms at presentation, primary care led investigations, and many others (16). 83% of participating surgeries completed over 95% of patient NCDA data with over 17,000 records submitted.'

b. How successful, and accurate, was the linkage between NCDA and RtD datasets? What was done with cases with multiple cancer diagnoses?

The NCDA and RtD datasets are tumour level data. They were linked by tumour id. There were 385 patients in the NCDA with more than one tumour. Where this occurred the GPs completed the primary care information relevant to that tumour, the patient's characteristics were the same, but the tumour information such as presenting symptoms and investigations were submitted to the NCDA by tumour. A sentence has been added to the Data Linkage section of the methods to explain this and reads as follows:

'Where a patient had multiple cancers (n=385), GPs were instructed to enter the same demographic and patient details, but submitted symptoms, investigations and interval data for each tumour separately.'

c. How was the date of first relevant presentation to a GP determined?

The GPs completing the NCDA were instructed to complete this question as follows:

'the time point at which, given the presenting signs, symptoms, history and other risk factors, it would be at least possible for the clinician seeing the patient to have started investigation or referral for possible important pathology, including cancer'. This follows the Aarhus statement which governs cancer diagnostic research (Ref 19 in paper – listed at the end of this document).

The date on which the patient first presented with symptoms ultimately attributed by the GP to the diagnosis of cancer. This is the date on which the patient presented at the Place of 1st presentation.

The text in the methods has been amended to:

'The primary care interval (PCI) was defined as the time from first relevant presentation to the GP to when the patient is referred into secondary care (19). The first presentation date in the NCDA was completed as the date when the patient first presented with symptoms ultimately attributed by the GP to the diagnosis of cancer (16).'

d. It was a little unclear how the MDC referral criteria and common presenting symptoms of MDCs were determined – what were the common presenting symptoms in MDCs vs referral criteria? Which MDCs were data taken to determine common symptoms / how many patients was this for? How was 'common' defined? Has this data been published?

The symptoms for the MDCs were determined within the ACE programme, following discussions with significant clinical engagement. This is published as part of a report published by the ACE programme. This has now been referenced in the paper (Reference 18). The common symptom also included was 'change in bowel habit', which is in the (still to be published) results paper of the MDCs. Previous work in Denmark (referred to in the Introduction, reference 4: Ingeman et al. 2015) was also an important source for defining the symptoms for the MDCs, they had trialled a referral pathway for patients with non-specific symptoms. We have added a reference for the MDC referral criteria of a recently published report and have updated the paragraph as follows:

'There were 84 distinct symptoms listed in the NCDA. To reflect patients being referred into the MDCs, the patients in the linked dataset were allocated to one of two groups depending on the symptoms at presentation to the GP within the NCDA dataset. The symptoms used to allocate patients were derived from the combined referral criteria to the MDCs (18) and an additional common presenting symptom (Bowel habit change) of the MDCs. These symptoms are listed in table 1.'

5. It appears that some significant effects are not highlighted in Table 4 (e.g. unadjusted OR for age 25-44, and 45-59; non-NSCS unadjusted OR for age 25-44, 45-59; unadjusted OR stage 2; non-NSCS unadjusted OR stage 2).

We have updated the tables reflecting the addition of site as a co-variate and double-checked that all significant ORs are bolded Where the confidence intervals have 1.00 as either the upper or lower CI, these are not bolded and not significant effects.

6. The abstract refers to the MDC-group vs non-MDC group, but paper uses terms NSCS and non-NSCS; consistency would be useful.

The abstract has been updated to NSCS/non-NSCS mirroring the main body of the paper for consistency.

7. Sample size for regression models are not stated and abbreviations not defined in results tables.

Abbreviations have now been defined in the tables, either within the tables or the table legends. The regression was undertaken on the full samples for each of the cohort, the entire NCDA and both NSCS/non-NSCS groups. The results have also been updated with the following statement to indicate the sample sizes of the regression models:

'Table S2 (Supplementary Information) shows unadjusted and adjusted odds ratios (ORs) of having a long PCI for the entire NCDA cohort (n=12,873) and stratified by NSCS (n=2,974) /non-NSCS (n=10,079) group.'

Reviewer: 2

Patients with NICE describe alarm symptoms for cancer are investigated via a rapid access (TWW) pathway. However, the majority of such patients do not have cancer and there is a serious question as to whether this pathway is disadvantaging patients with less specific symptoms. With some cancers, it is clear that a majority of cancer presentations are from other than the TWW patients. How to manage this group is a significant issue in cancer care as they are clearly more numerous.

In this study the authors try and use cancer data linked to primary care data to establish the characteristics of a cohort similar to those who may be seen in one of the experimental Multidisciplinary Diagnostic Centres (MDCs) in which the authors have an interest.

The results reveal that this cohort with non-specific symptoms were likely to be older, more deprived, more comorbid and have higher stage disease compared to the “other group”. The results as far as they go are of some interest although largely predictable.

This study provides evidence adding to anecdotal information reported by patients/clinicians of difficulties facing patients who present with non-specific symptoms. It also provides a case for change for those interested in influencing policy.

It describes the characteristics of a group who had ultimately a diagnosis of cancer but not the denominator. The denominator is critical if the utility of MDCs is to be assessed.

This paper is not assessing the utility of the MDCs; instead it focuses on using linked data to compare patients with different symptom profiles and provides evidence of diagnostic pathway issues for patients similar to those going through the MDCs. We can only analyse the data we have; data for those without cancer is not available. A paper evaluating the MDCs is being published separately and this is included the discussion.

There are some specific problems with this paper.

1. It is very difficult to follow bearing in mind the relatively simple (I think) message. It is also over long.

Our article at 2,692 words is significantly lower than the recommended maximum word limit by BMJ Open (4,000 words). We have standardised the abbreviations and have tried to keep this amended paper as simple as possible whilst reflecting our methods and results. We have also clarified the results by defining acronyms and moving the regression results table to the Supplementary Information, which hopefully improves the readability.

2. The numerous similar abbreviations do little enhance it accessibly.

We have changed abbreviations in the abstract to make them consistent throughout the paper. There has been much discussion within the authorship team, including with clinicians, to define the terminology. We decided on non-specific but concerning symptoms (NSCS) to reflect as much as possible what our study and the MDCs focus on.

3. Although I have read it many time I still don't know if non-NSCS is the TWW group. If it is please say so. If it isn't could the authors explain.

This paper is entirely driven by the symptoms that prompt MDC referral and is not a TWW/non-TWW comparison. A list of TWW/non-TWW symptoms is not readily available; instead our work aims to describe experiences of diagnostic pathways for patients similar to those referred into the MDCs. In the planning stages of this project, a TWW/non-TWW comparison was suggested. There are several symptoms in the MDC referral list that have numerous recommendations, some of which are TWW referrals. It was decided that this work should focus on describing and comparing patients similar to the MDC referred. Our analysis included a comparison of proportions diagnosed via different routes and we confirmed that patients in the non-NSCS group are more likely to be referred on the TWW route – there are higher proportions diagnosed via this route in the non-NSCS group (47%) compared with the NSCS group (29%).

We have amended a paragraph in the discussion which explains what our analysis does not do – namely, compare TWW with non-TWW symptoms, as follows:

‘The allocation into the NSCS and non-NSCS groups by symptom is a proxy for distinguishing between alarm and non-alarm symptoms in the NICE referral guidelines. An analysis of symptom groups with a true separation between alarm symptoms warranting a referral onto an urgent referral pathway and vague symptoms which do not, would be very difficult. This is especially the case for

symptoms where there are more than one recommendation depending on other symptoms and patient characteristics (such as appetite loss, with five different recommendations and weight loss with 13 recommendations). Such analysis would also require patient characteristic information which is not all available in the NCDA linked data. This study, however, focused on the MDC referral criteria and common presenting symptoms recorded at MDCs. Indeed, the aim of this work was to provide evidence of the possible diagnostic problems facing patients similar to those potentially eligible to go through the MDCs'.

4. There is a large amount of data in the tables. I doubt anybody apart from a hard-core epidemiologist will actually take an interest in these data and perhaps much could be a as supplement.

We have moved the regression results table to the Supplementary Information file. However, we also hope that some hard-core epidemiologist will value this work (!), so our relegation of the regression to the Supplementary Information file is targeted at maximising the readability for the soft-core reader.

5. The important data could be presented more simply.

We have moved the regression results to the Supplementary Information file; hopefully this has simplified our results. As always, there is a tension between completeness and simplicity; arguably we have veered to completeness. This is safer than the alternative. Readers can choose to omit what they consider irrelevant: it's impossible to read what's not there!

6. Although the limitations of the study are presented the reality is this simply describes characteristic of patients who turned out to have cancer but did not have (I think) TWW symptoms and it does not crucially look at outcome.

Our objective was to not look at outcomes, instead we focused on describing differences between patients similar to those referred onto the MDCs and those not. We wanted to focus on diagnostic pathways of cancer patients who present with symptoms potentially leading to elongated and complex pathways. Patients who present with non-specific symptoms have less clear referral pathways under the TWW system. When a patient presents to a GP with non-specific symptoms the GP doesn't necessarily know which pathway/referral speciality to refer into – whereas with the TWW system, the GP needs to have an idea. This work is mainly descriptive in nature, but provides important evidence describing differences between groups of patients with symptom differences. This can provide the basis for considering alternative referral routes for different groups of patients.

7. If MDCs were going to be promoted more evidence would be needed. It is as yet unclear that the TWW system has utility. Whilst this is not the aim of the paper some discussion on how it might impact on future studies would be of value

We wholly agree that this paper alone would not suffice to evaluate (not promote – we're scientists!) MDCs per se. This paper does not focus on evaluating the MDCs nor assessing the utility of the TWW system. Results of the MDC evaluations are being published separately and we have added a sentence saying this to the discussion. What this paper provides is context for the MDCs on patients presenting with symptom profiles the same as those going through the MDCs, demonstrating that patients with non-specific symptoms have longer intervals. Future analysis of the NCDA could help assess utility of the TWW system.

The highlights section of the paper also describes the distinction between our paper and any TWW/non-TWW symptom comparison

8. Although the authors do comment on jaundice not being in any sense being a non specific symptom, it is so completely ridiculous that these data should probably be excluded.

The inclusion of jaundice as an MDC referral symptom was discussed at length within the project team, alongside clinical input. The fundamental issue is that jaundice is included in the entry criteria for some MDCs (and we wholly agree that was a debatable decision by them). Therefore if we want to mirror MDCs, we have to include jaundice. We have retained the symptom list as it is, but conducted a sensitivity analysis removing it. The sensitivity analysis is little different, fortunately, making this point ultimately rather academic.

Reference

19. Weller D, Vedsted P, Rubin G, Walter FM, Emery J, Scott S, et al. The Aarhus statement: improving design and reporting of studies on early cancer diagnosis. *Br J Cancer*. 2012;106(7):1262-7.

VERSION 2 – REVIEW

REVIEWER	Rebecca Bergin Cancer Council Victoria, Australia
REVIEW RETURNED	11-Oct-2019
GENERAL COMMENTS	Happy with this response, a very nice paper.